# DArTseq-based silicoDArT and SNP markers reveal the genetic diversity and population structure of Kenyan cashew (*Anacardium occidentale* L.) landraces

**Dennis Wamalabe Mukhebi**[1], **Pauline Wambui Gachanja**[1], **Diana Jepkoech Karan**[2], **Brenda Muthoni Kamau**[1], **Pauline Wangeci King'ori**[1], **Bicko Steve Juma**[2,3], **Wilton Mwema Mbinda**[1,2]*

**1** Department of Biochemistry and Biotechnology, Pwani University, Kilifi, Kenya, **2** Pwani University Bioscience Research Center (PUBReC), Pwani University, Kilifi, Kenya, **3** Department of Agroecology, Crop Genetics and Biotechnology Section, Aarhus University, Flakkebjerg, Denmark

* wilton.mbinda@gmail.com, w.mbinda@pu.ac.ke

## Abstract

Cashew (*Anacardium occidentale* L.) is an important tree grown worldwide for its edible fruits, nuts and other products of industrial applications. The ecologically sensitive cashew-growing region in coastal Kenya is significantly affected by rising temperatures, droughts, floods, and shifting rainfall patterns. These changes adversely impact cashew growth by altering flowering patterns, increasing pests and diseases, and causing post-harvest losses, which ultimately result in reduced yields and tree mortality. This is exacerbated by the long juvenile phase, high heterozygosity, lack of trait correlations, large mature plant size, and inadequate genomic resources. For the first time, the Diversity Array Technology (DArT) technology was employed to identify DArT (silicoDArT) and single nucleotide polymorphisms (SNPs) markers for genomic understanding of cashew in Kenya. Cashew leaf samples were collected in Kwale, Kilifi and Lamu counties along coastal Kenya followed by DNA extraction. The reduced libraries were sequenced using Hiseq 2500 Illumina sequencer, and the SNPs called using DarTsoft14. A total of 27,495 silicoDArT and 17,008 SNP markers were reported, of which 1340 silicoDArT and 824 SNP markers were used for analyses after screening, with >80% call rate, >95% reproducibility, polymorphism information content (PIC ≥ 0.25) and one ratio (>0.25). The silicoDArT and SNP markers had mean PIC values ranging from 0.02–0.50 and 0.0–0.5, with an allelic richness ranging from 1.992 to 1.994 for silicoDArT and 1.862 to 1.889 for SNP markers. The observed heterozygosity and expected values ranged from 0.50–0.55 and 0.34–0.37, and 0.56–0.57 and 0.33 for both silicoDArT and SNP markers respectively. Understanding cashew genomics through the application of SilicoDArT and SNP markers is crucial for advancing cashew genomic breeding programs aimed at improving yield and nut quality, and enhancing resistance or tolerance to biotic and abiotic stresses. Our study presents an overview of the genetic diversity of cashew landraces in Kenya and demonstrates that DArT systems are a reliable tool for advancing genomic research in cashew breeding.

**Data availability statement:** All relevant data are within the Supporting information files

**Funding:** This research was supported by the National Research Fund, Kenya (NRF/2/MMC/158). The funders had no role in the design of the study; in the collection, analyses, or interpretation of data; in the writing of the manuscript; or in the decision to publish the results.

**Competing interests:** The authors have declared that no competing interests exist.

## Introduction

Cashew is a tropical evergreen tree from the *Anacardiaceae* family, which comprises approximately 75 genera and 700 species. Its origin is traced from the northeastern region of Brazil [1]. In Africa, Portuguese traders first introduced cashew in Mozambique in the 16th century for land reclamation [2]. From Mozambique, cashew was spread to East Africa [3]. Africa accounts for more than 50% of the global production of cashew nuts, making the crop an important source of income for smallholder farmers and the regional economy [4]. The evergreen perennial tropical tree is cultivated primarily for its nuts, the second most important edible tree nuts after almond nuts [5]. Other cashew products include the cashew apple (fleshy pseudo fruit), gum, wood, kernel oils, and nutshell liquid [6] which are utilized in food and other industries, making the crop an important source of income for small-scale farmers and the national economy. Cashew trees play key biodiversity roles as sources of food and shelter for wildlife. Additionally, the cashew tree deep rooting system supports the provision of ecosystem services such as soil stabilization, erosion prevention and carbon sequestration [7]. This is a potential adaptation strategy to suboptimal or adverse environmental conditions such as extreme salinity and poor drainage, suggesting that its integration into agroforestry systems makes it suitable for upscaling climate-smart technologies and building resilience to climate change among smallholder farmers.

*A. occidentale* is a diploid (2n = 42) self-pollinated plant with an approximate genome size of 420 Mb [8]. The intrinsic heterozygosity due to the plant's predominant allogamy makes the cashew genome complex [9], causing a major hindrance in genomics studies and genetic improvement [5]. This challenge is also compounded by the plant's long juvenile phase, lack of juvenile–mature trait correlations and large size of the mature plant [10], making the plant's conventional breeding strategies slow and unpredictable. Additionally, climate change, which is causing a rise in temperatures, extended droughts, floods, and shifts in rainfall patterns, persists in disrupting cashew farming through alterations in flowering, fruiting, pests and diseases occurrences, wilting of plants inundated farms, and postharvest losses [11]. Genomic techniques should be applied in cashew breeding programs to reduce the errors of conventional breeding approaches and enhance breeding efficiency.

Highly efficient, robust, and polymorphic molecular markers are essential for genetic analysis and marker-assisted breeding in plants. In the past, several markers, including random amplified polymorphic DNA [12–14], simple sequence repeats [13,15] and amplified fragment length polymorphisms [15] have been utilized. These are classified as dominant markers and were later superseded by codominant markers such as simple sequence repeats, single nucleotide polymorphisms (SNPs) and diversity array technology (DArT), which are more informative and produce consistent results than dominant markers [16]. Until recently, cashew genomic resources were characterized using microsatellites [17] and insertion/deletion (InDel) markers [10]. The paucity of effective molecular markers and the absence of cashew genomic sequence data have hampered genetic and genomics studies, and molecular breeding.

DArT is a genotyping-by-sequencing platform that utilizes restriction enzymes to digest an organism's genome, followed by high-throughput sequencing of the fragments to discover genome-wide markers. The choice of restriction enzymes used permits the segregation of highly informative low-copy fragments of the genome. DArTseq-based analysis generates silicoDArT markers (presence/absence of restriction fragments in representation), which are dominant, and single-nucleotide polymorphism (SNP) markers, which are codominant [18]. This platform is widely used in genomic research due to its efficiency in generating a large number of molecular markers across the entire genome in a short time. It is also easy to use and is relatively cost-effective [19]. DArT technology has been applied to several crops, including barley [20], cassava [21], maize [22], macadamia [23,24], almond

[25], and cashew [26]. However, there are limited reports on the application of DArTseq markers in cashew or any genomic studies on Kenyan cashew breeding programs. In the present study, we report for the first time, the effectiveness of DArTseq-based silicoDArT and SNP markers as alternative markers to study the genetic diversity and population structure among a collection of cashew landraces along the Kenyan coast. The silicoDArT and SNP resources generated here lay a foundation for future genome-wide association studies or genomic selection in cashew.

## Materials and methods

### Plant material and DNA extraction

Leaf samples from field-grown cashew trees were collected from farmers in Kwale, Kilifi and Lamu Counties via a stratified random system as described by [27]. The farmers were grouped into strata that were equivalent to the sub-counties. The number of samples collected was proportional to the number of farmers in the stratum, and the density of cashew trees in each sub-county (Table 1). All the plant materials were landraces, and 92 apical leaf samples (27 from Kwale, 28 Lamu, and 37 Kilifi) were collected. The samples were transferred to Falcon tubes containing 20% DESS (a solution containing dimethyl sulfoxide, disodium EDTA, and saturated NaCl buffer [28] and transported in thermally insulated boxes to the Pwani University Bioscience Research Centre (PUBReC) following standard laboratory operating protocols. The collected samples were used for high-quality genomic DNA (gDNA) isolation using Qiagen Plant Mini kit (Hilden, Germany). The quantity and integrity of the extracted gDNA were evaluated using a Nanodrop 2000c spectrophotometer and 1% agarose gel electrophoresis, respectively. The gDNA was then shipped to SEQART AFRICA, Nairobi, Kenya, for genotyping.

### DArTseq genotyping and marker quality control

Diversity Array Technology libraries were constructed according to [29]. The DArT-seq complexity reduction method was employed by digesting gDNA with two restriction enzymes (*PstI* and *MseI*) and ligating barcoded adapters, followed by PCR amplification of adapter-ligated fragments. Pooling and purification of the PCR amplicons were performed with a QIAquick PCR purification kit (QIAGEN Hilden, Germany). A single-read Illumina HiSeq 2500 platform for 77 cycles was used to sequence purified PCR products. DarTsoft14, an in-house marker-scoring program was used to score the DArTseq markers (S1–S3 Tables).

Two types of DArT markers were scored: SilicoDArT and SNP markers. Both markers were scored as binary '1' for the presence or '0' for the absence of a marker in the genomic representation of each sample as described by [30]. Markers were tested for call rate (%), one ratio, and reproducibility (%, a proportion of technical replicate assay pairs in which the marker score is consistent) [23]. Both silicoDArT and SNP markers were filtered based on a minor call rate (≥80%), one ratio (>0.25) and reproducibility (≥95%).

### Genetic diversity and population structure analysis

Genetic diversity analyses were conducted to determine the polymorphic information content (PIC), observed heterozygosity (Ho), expected heterozygosity (He), allelic richness, and fixation indices (pairwise and overall FST). The genetic indices were computed in R statistical software using adegenet package. A landscape and ecological associations (LEA) R package v.3.19 was used to determine the population genetic structure among the cashew samples across Kwale, Kilifi and Lamu counties. Using the *snmf* function of LEA, ancestry coefficients

**Table 1. Cashew landrace collections across Kwale, Kilifi and Lamu counties in coastal Kenya.**

| County | Sub-county | Cashew landraces |
| --- | --- | --- |
| Kwale | Msambweni | KWL/MS/01 |
| Lamu | Mkunumbi | LM/LW/94 |
| Lamu | Mpeketoni | LM/LW/104 |
| Lamu | Mkowe | LM/LE/113 |
| Kwale | Msambweni | KWL/MS/10 |
| Kwale | Lungalunga | KWL/LU/21 |
| Kwale | Matuga | KWL/MAT/30 |
| Kilifi | Magarini | KLF/MAL/41 |
| Kilifi | Malindi | KLF/MAL/52 |
| Kilifi | Kaloleni | KLF/KAL/63 |
| Kilifi | Kilifi South | KLF/KS/74 |
| Lamu | Hongwe | LM/LW/86 |
| Kwale | Msambweni | KWL/MS/02 |
| Lamu | Witu | LM/LW/96 |
| Lamu | Hindi | LM/LW/105 |
| Lamu | Mkowe | LM/LE/114 |
| Kwale | Msambweni | KWL/MS/12 |
| Kwale | Matuga | KWL/MAT/32 |
| Kilifi | Magarini | KLF/MAL/42 |
| Kilifi | Malindi | KLF/MAL/54 |
| Kilifi | Kaloleni | KLF/KAL/64 |
| Kilifi | Kilifi South | KLF/KS/75 |
| Lamu | Hongwe | LM/LW/87 |
| Kwale | Msambweni | KWL/MS/03 |
| Lamu | Witu | LM/LW/97 |
| Lamu | Hindi | LM/LW/106 |
| Lamu | Bahari | LM/LE/115 |
| Kwale | Msambweni | KWL/MS/14 |
| Kwale | Lungalunga | KWL/LU/24 |
| Kwale | Matuga | KWL/MAT/33 |
| Kilifi | Magarini | KLF/MAL/44 |
| Kilifi | Malindi | KLF/MAL/56 |
| Kilifi | Kaloleni | KLF/KAL/66 |
| Kilifi | Kilifi North | KLF/KN/77 |
| Lamu | Hongwe | LM/LW/88 |
| Kwale | Msambweni | KWL/MS/04 |
| Lamu | Witu | LM/LW/98 |
| Lamu | HindI | LM/LW/107 |
| Lamu | Bahari | LM/LE/117 |
| Kwale | Msambweni | KWL/MS/15 |
| Kwale | Matuga | KWL/MAT/25 |
| Kwale | Matuga | KWL/MAT/34 |
| Kilifi | Magarini | KLF/MAL/45 |
| Kilifi | Malindi | KLF/MAL/57 |
| Kilifi | Kaloleni | KLF/KAL/67 |
| Kilifi | Kilifi North | KLF/KN/79 |
| Lamu | Hongwe | LM/LW/89 |

*(Continued)*

**Table 1.** (Continued)

| County | Sub-county | Cashew landraces |
|---|---|---|
| Kwale | Msambweni | KWL/MS/05 |
| Lamu | Mapenya | LM/LW/100 |
| Lamu | Hindi | LM/LW/109 |
| Lamu | Bahari | LM/LE/118 |
| Kwale | Lungalunga | KWL/LU/16 |
| Kwale | Matuga | KWL/MAT/35 |
| Kilifi | Magarini | KLF/MAL/46 |
| Kilifi | Malindi | KLF/MAL/58 |
| Kilifi | Kaloleni | KLF/KAL/69 |
| Kilifi | Kilifi North | KLF/KN/80 |
| Lamu | Mkunumbi | LM/LW/90 |
| Kwale | Msambweni | KWL/MS/06 |
| Lamu | Uziwa | LM/LW/101 |
| Lamu | Dabo | LM/LE/110 |
| Lamu | Tewe | LM/LE/119 |
| Kwale | Lungalunga | KWL/LU/17 |
| Kwale | Matuga | KWL/MAT/27 |
| Kwale | Kinango | KWL/KN/38 |
| Kilifi | Magarini | KLF/MAL/48 |
| Kilifi | Malindi | KLF/MAL/59 |
| Kilifi | Kilifi South | KLF/KS/70 |
| Kilifi | Kilifi North | KLF/KN/81 |
| Lamu | Mkunumbi | LM/LW/91 |
| Kwale | Msambweni | KWL/MS/07 |
| Lamu | Uziwa | LM/LW/102 |
| Lamu | Langoni | LM/LE/111 |
| Kwale | Lungalunga | KWL/LU/18 |
| Kwale | Matuga | KWL/MAT/28 |
| Kwale | Kinango | KWL/KN/39 |
| Kilifi | Magarini | KLF/MAL/49 |
| Kilifi | Ganze | KLF/GAN/60 |
| Kilifi | Kilifi South | KLF/KS/72 |
| Kilifi | Kilifi North | KLF/KN/83 |
| Lamu | Mkunumbi | LM/LW/92 |
| Kwale | Msambweni | KWL/MS/08 |
| Lamu | Mpeketoni | LM/LW/103 |
| Lamu | Langoni | LM/LE/112 |
| Kwale | Lungalunga | KWL/LU/20 |
| Kwale | Matuga | KWL/MAT/29 |
| Kwale | Kinango | KWL/KN/40 |
| Kilifi | Malindi | KLF/MAL/51 |
| Kilifi | Kaloleni | KLF/KAL/61 |
| Kilifi | Kilifi South | KLF/KS/73 |
| Kilifi | Kilifi North | KLF/KN/84/ |
| Lamu | Mkunumbi | LM/LW/93 |

were estimated where cross-entropy criteria were used to identify the optimum number of k populations where k = 1–10 with 10 repetitions. Cashew landraces in the subpopulations of Kwale, Kilifi and Lamu counties were identified in R statistical software using Q matrix. Admixture plots of individual cashew landraces were plotted with the *plot* function of base R. The results of individual markers (silicoDArT and SNP markers) from cashew landraces obtained from population analysis were subjected to analysis of molecular variance (AMOVA) using the poppr package and principal coordinate analysis (PCoA) in R statistical software. Weighted neighbor-joining dendrograms were constructed for both markers.

## Results

### Marker characterization

A total of 27,494 silicoDArT and 17,008 SNP markers from 92 apical leaves of cashew landraces were filtered, resulting in 1340 and 824 silicoDArt and SNP markers, respectively. All the identified markers had reproducibility ≥95% (Fig 1A and B) and a call rate of ≥80% (Fig 1C and D).

### Genetic relationships among cashew landraces

Among the 1340 and 824 informative markers (silicoDArt and SNP), the observed PIC values varied across the markers between 0.15 and 0.30 and between 0.45 and 0.50 (Fig 2). The silicoDArT and SNP markers presented PIC range values of 0.02–0.50 and 0.0–0.50 and average of 0.29 and 0.24, respectively (Table 2). The allelic richness among the populations ranged from 1.992 to 1.994 for silicoDArT and from 1.862 to 1.889 for the SNP markers (Table 3). The

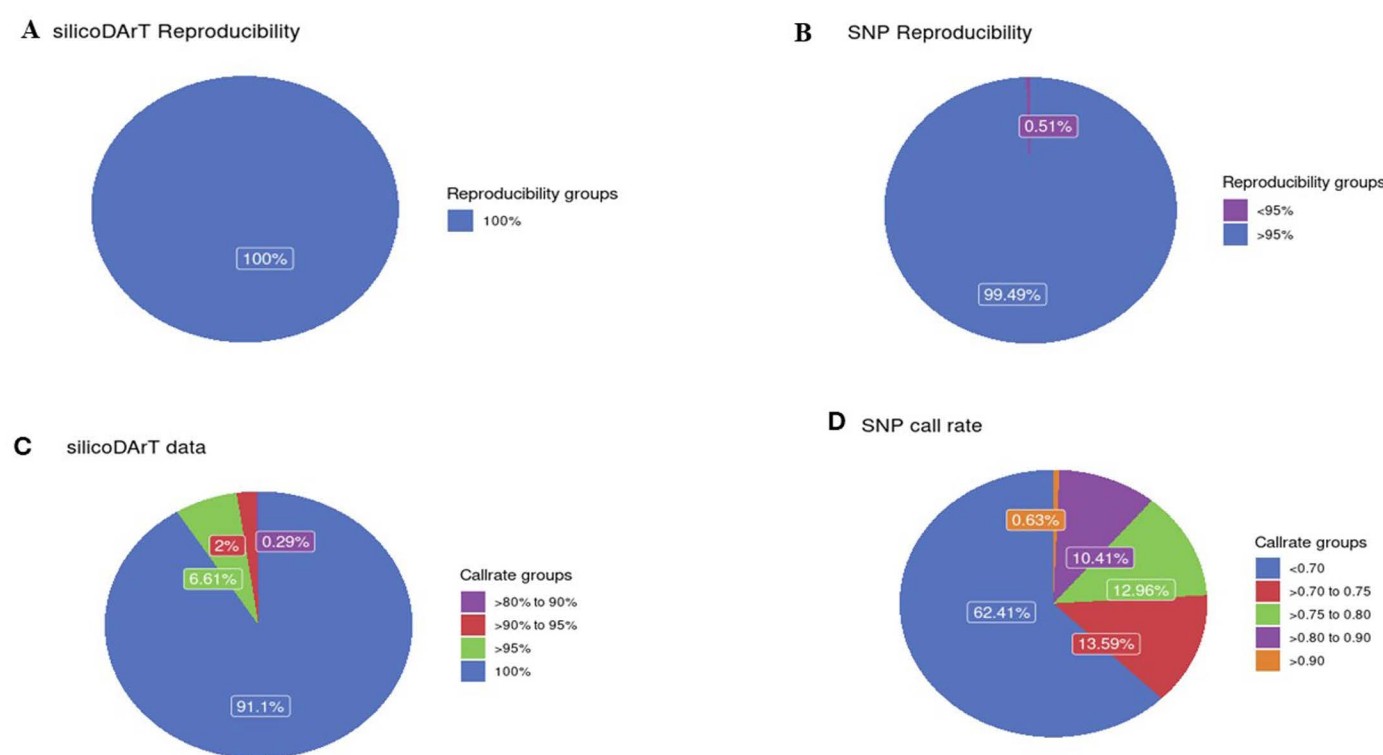

**Fig 1. Distribution of marker data for reproducibility and call rate. A.** Reproducibility of SilicoDArT markers, **B.** Reproducibility of SNP markers. **C.** Call rates of SilicoDArT markers, **D.** Call rates of SNP markers.

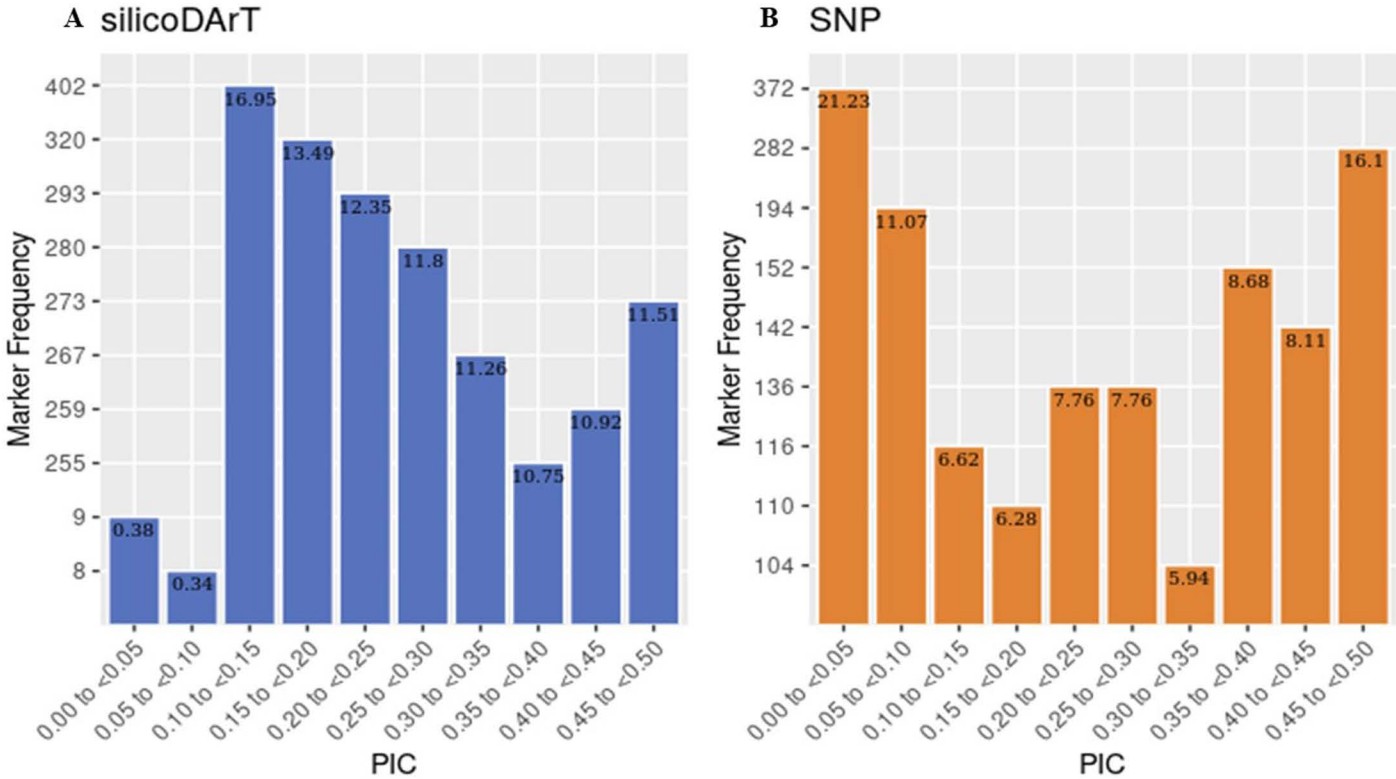

**Fig 2. Distribution of PIC values of markers used for genomic studies in 92 cashew landrace genotypes. A.** silicoDArT markers and **B.** SNP markers.

**Table 2. Polymorphic distribution for both silicoDArT and SNP markers.**

| Parameter | silicoDArT | SNP |
|---|---|---|
| Mean | 0.29 | 0.24 |
| Median | 0.28 | 0.23 |
| Range | 0.02 to o.50 | 0.0 to 0.5 |

**Table 3. Statistics of the genetic diversity indices for SilicoDArT and SNP markers from cashew landraces along the Kenyan coast.**

| Location | Sub-population | SilicoDArT | | | SNP | | |
|---|---|---|---|---|---|---|---|
| | | He | Ho | Ar | He | Ho | Ar |
| Kilifi | 32 | 0.340 | 0.500 | 1.992 | 0.330 | 0.560 | 1.888 |
| Kwale | 30 | 0.350 | 0.510 | 1.994 | 0.330 | 0.560 | 1.889 |
| Lamu | 30 | 0.370 | 0.550 | 1.994 | 0.330 | 0.570 | 1.862 |

observed heterozygosity and expected values for the silicoDArT markers ranged from 0.50–0.55 and 0.34–0.37, while the SNPs ranged from 0.56–0.57 and 0.33, respectively (Table 3).

Population structure analysis using cross-entropy criteria revealed that individual cashew landraces could be divided into two subpopulations (K = 2) for both markers (Fig 3A and 3B). These findings were validated by visualizing the admixture bar plot of the Q matrix (Figs 4 and 5). The overall fixation indices among the subpopulations for the silicoDArT and

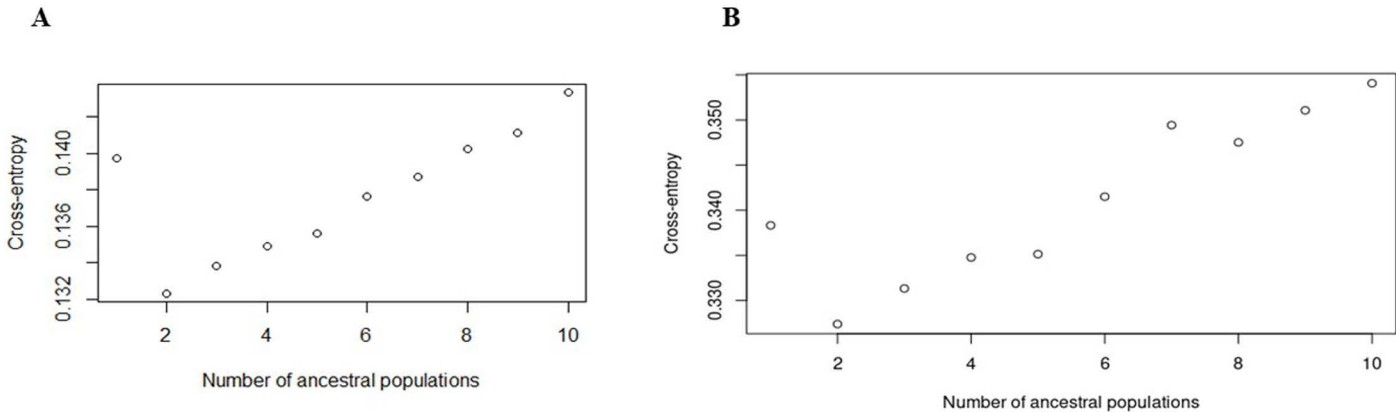

**Fig 3. Cashew landrace population structure determination based cross-entropy of Q Matrix for population admixture. A**. SilicoDArT markers, **B.** SNP markers.

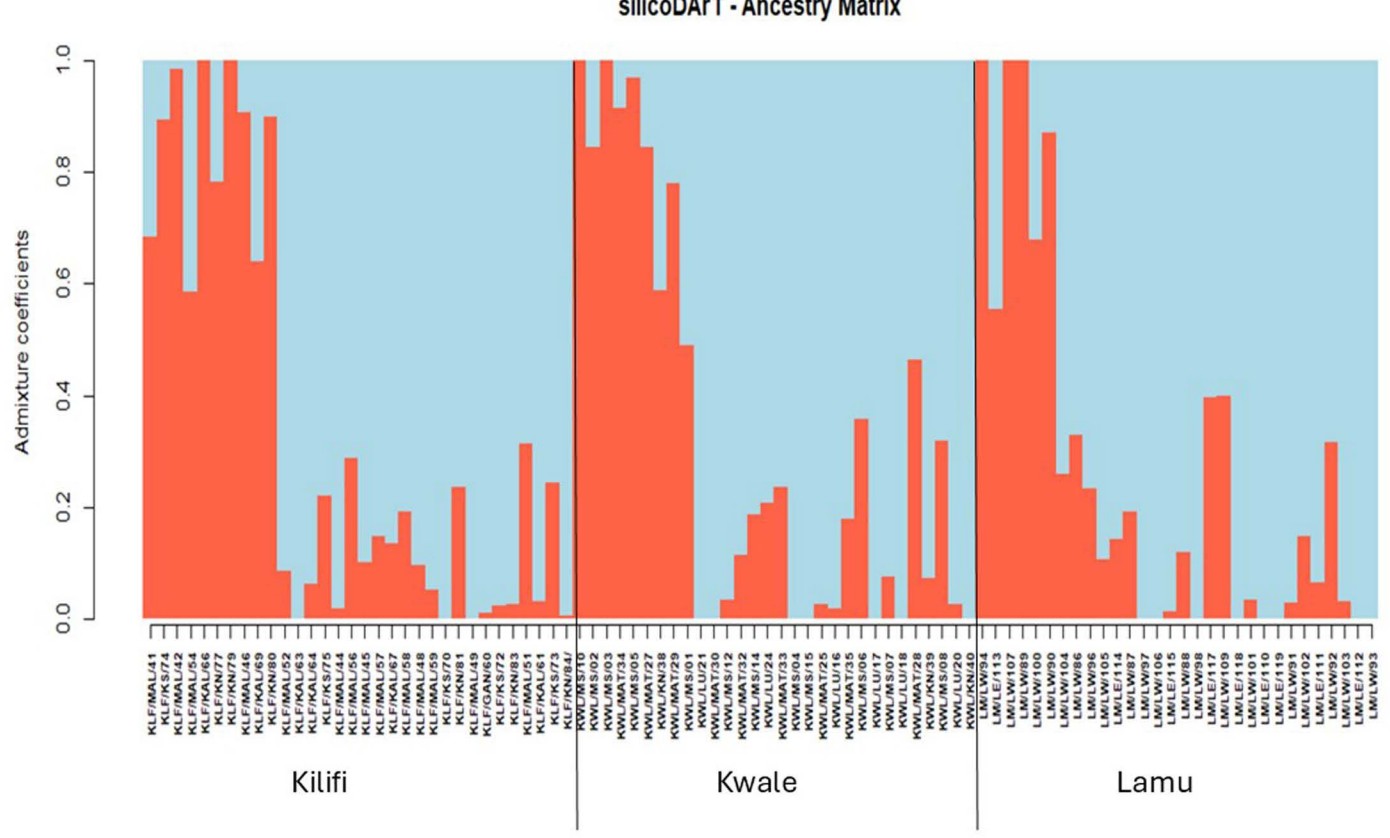

**Fig 4. Population admixtures of 92 cashew landrace genotypes based on the Q matrix derived from 1,340 high-density silicoDArT markers.**

SNP markers were 0.003 and 0.008, respectively (Table 4). The extent of genetic differentiation across the three study sites was substantially low (Table 4) as indicated by the low overall fixation indices among the subpopulations for both the silicoDArT (Fst = 0.003) and SNP (Fst = 0.008) markers (Table 4). Analysis of molecular variance revealed a 6.8% variation between

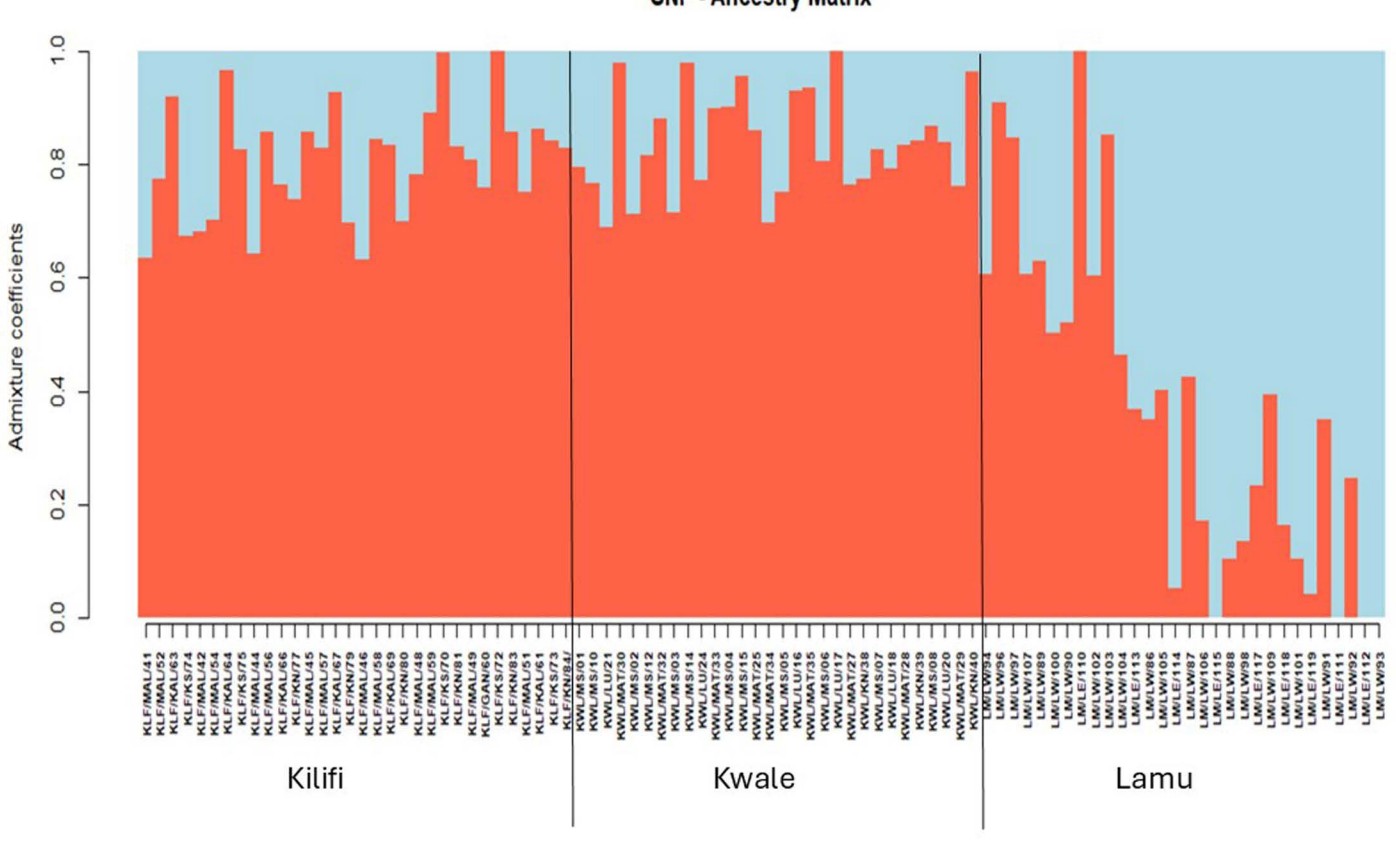

**Fig 5. Population admixtures of 92 cashew landrace genotypes based on the Q matrix for 824 silicoDArT markers.**

**Table 4. Population pair-wise fixation index of cashew landrace genotypes genotyped using silico DArT and SNPs markers.**

| SilicoDArT Overall FST 0.003019035 | | | SNP Overall FST 0.008888716 | | |
|---|---|---|---|---|---|
| | Kwale | Lamu | | Kwale | Lamu |
| Lamu | 0.005 | | Lamu | 0.0129 | |
| Kilifi | −0.0011 | 0.0051 | Kilifi | 0.0027 | 0.0111 |

**Table 5. Analysis of molecular variance in cashew landrace genotype collection based silicoDArT and SNP markers.**

| Source of variation | DF | SS | MS | %Variance |
|---|---|---|---|---|
| Between population | 2 | 425.097 | 212.548 | 6.80 |
| Within population | 89 | 5845.872 | 65.684 | 93.20 |
| Total | 91 | 6270.969 | 68.112 | 100 |

DF: Degree of freedom, SS: Sum of squares, MS: Mean squares.

the cashew landrace populations and 93.2% within them (Table 5). Further evaluation with a principal coordinate analysis (PCoA) revealed that most cashew landraces in Kilifi and Kwale counties share a genetic lineage, as compared to that of Lamu on the PCoA plot for both silicoDArT and SNP markers (Fig 6). Weighted neighbor‑joining hierarchical clustering assorted

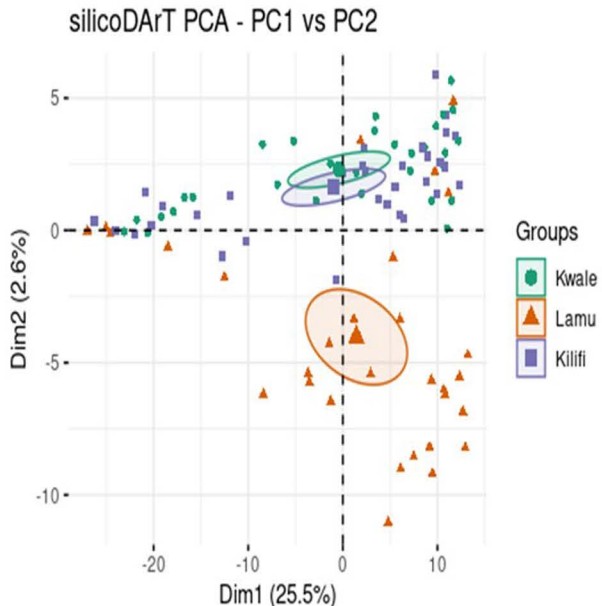
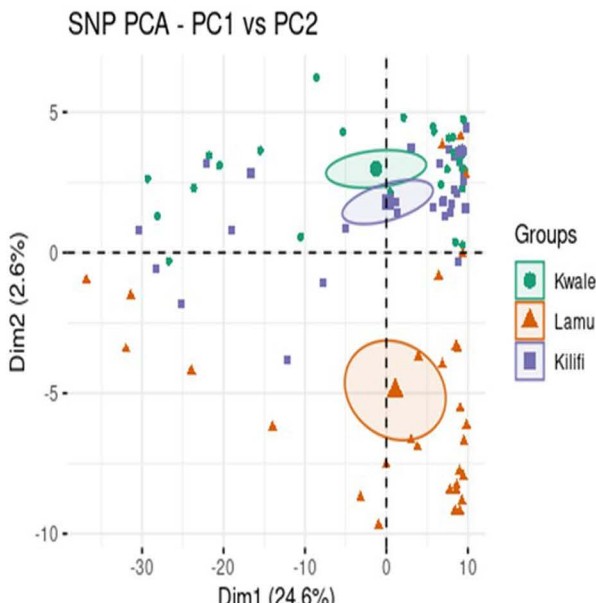

**Fig 6. Principal coordinate analysis of cashew landraces based on population structure via silicoDArT and SNP markers.**

the cashew landrace population into three clusters with different numbers of landraces: cluster 1 (53), cluster 2 (22) and cluster 3 (17) for SNP markers and cluster 1 (28), cluster 2 (39) and cluster 3 (25) for silicoDArT (Fig 7A). However, some landraces for SNP markers in cluster 2, such as LM/LW/103 and KLF/KAL/69, were clustered in cluster 1. The silicoDArT markers LM/LW/106, LM/LW/88 and LM/LE/114 of cluster 2 were clustered in cluster 3, whereas the KLF/MAL/52 and KWL/MAT/32 cashew landraces of cluster 3 were in cluster 2 (Fig 7B).

## Discussion

Establishing efficient and well-characterized genetic resources is a primary goal for successful breeding programs. This approach is fundamental for crop genetic resource management, breeding programs and understanding the ancestral relationships of genotypes [31]. Crop genetic of yield, quality, and response to biotic and abiotic stress resistance/tolerance depend on the efficiency of selection. However, few genomic resources utilizing random amplified polymorphic DNA, and simple sequence repeat markers have been developed for cashew varieties to support genetic differentiation and selection. Germplasm characterization of cashew genotypes has been performed through conventional molecular markers, yet they are few and currently only 51 sequences are available in the public database. We evaluated the effectiveness of DArTseq-based silicoDArT and SNP markers as alternative markers to analyze genetic diversity and population structure.

After filtering, 1,340 high-quality silicoDArT markers and 824 SNP markers were identified from the initial 27,494 silicoDArT and 17,008 SNP markers, which are recommended for use in genomic analysis [24]. Our analysis identified SNP and silicoDArT markers with PIC values ranging between 0.0 and 0.5, indicating a moderate genetic variation. The moderate genetic divergence of individual cashew landraces in this study corroborates the findings of [23] on macadamia, who reported PIC values of silicoDArT and SNP markers with moderate levels of informativeness. This low genetic diversity among the cashew landraces is strongly attributed to the local domestication and distribution of the crop in the Kenyan coastal

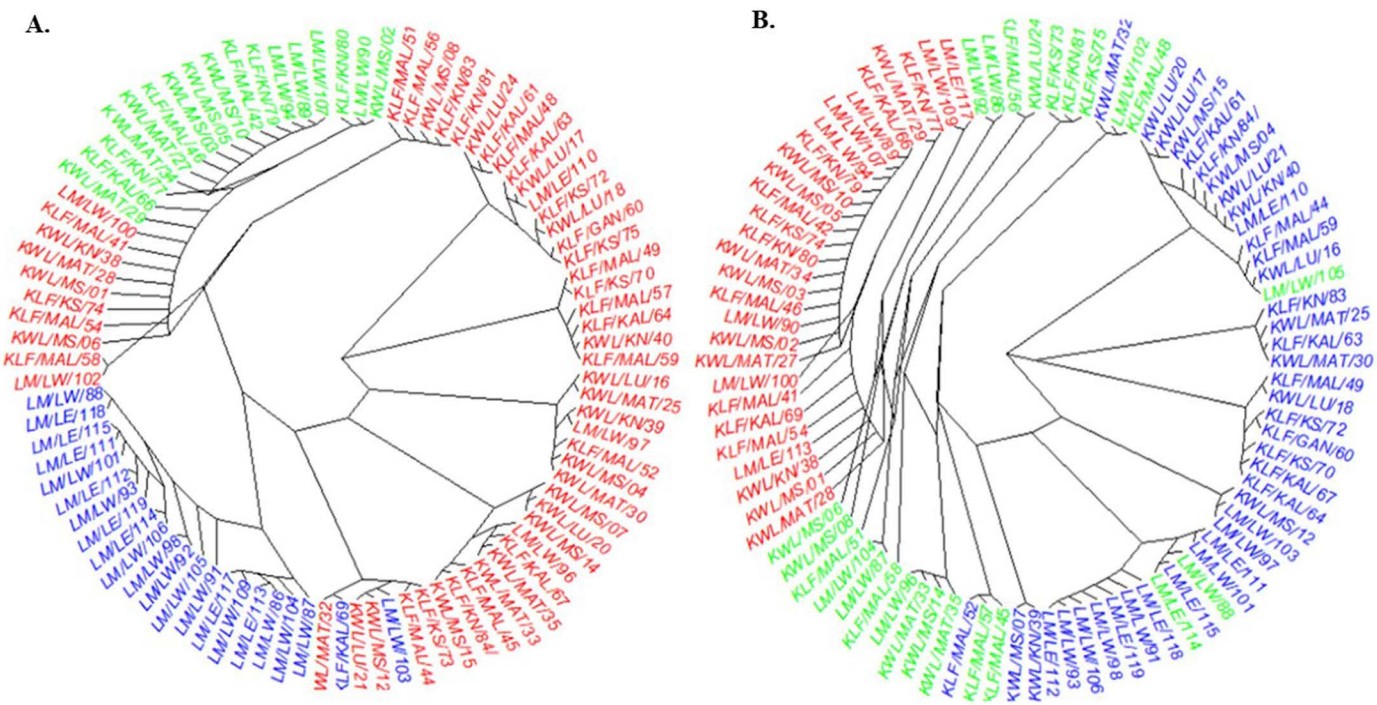

**Fig 7. Genetic clusters (red, blue and green clusters) among 92 cashew landrace genotypes based on A. silicoDArT and B. SNP markers.**

counties (Kwale, Kilifi and Lamu). The genetic diversity observed among the cashew landraces suggests a consistent gene flow across the populations.

The observed and expected heterozygosity were distinct, with observed heterozygosity being greater than expected. This observation could be attributed to outcrossing, reflecting a deviation from Hardy-Weinberg equilibrium in the analyzed cashew landraces. The outcrossing among the cashew landraces confers its fitness and relevance to withstand disease and enhanced environmental adaptability [32]. Cross-pollination between staminate and pistillate flowers enables cashew trees to predominantly outcross, which maintains certain alleles in the population, even if they might be detrimental in a homozygous state [33].

Both markers exhibited moderate allelic richness, ranging from 1.862 to 1.994 (Table 3), reflecting genetic drift among the individual cashew landraces. The observed moderate allelic richness among the cashew landraces explains their ability to adapt to diseases and climate changes [34,35]. The allelic richness observed in this study backtraces on the impact of outcrossing, as indicated by Ho> He, which contributes to the resilience of cashew trees to diseases and their adaptability to climate change. Our findings concur with those of [9], who reported average allelic richness alleles per locus among cashews in East Timor. The allelic richness among cashews reflects their evolutionary potential in breeding and conservation programs in East Africa and Kenya.

The presence of both dominant and recessive alleles among cashew landraces has important implications for selection and breeding. More often, the upregulation of dominant alleles may enhance the selection of well-adapted genotypes. However, this may hinder the selection of recessive alleles [36]. The fixation index values (Table 4) suggest limited gene exchange between populations, indicating a high flow of genetic information within cashew landraces. This lack of distinct population structure is evident in both the admixture barplot and the

PCoA plot. To maximize genetic gains when breeding for new cashew varieties, it is advisable to cross genetically divergent genotypes from different clusters. This approach can improve economic traits such as yield, nutrient composition, and resistance to heat, drought, and diseases.

Our study provides insight on the population structure, where two clusters (k = 2) were identified. This foundational work was essential in advancing the understanding of genetic diversity and population structure among 92 distinct cashew landraces from Kwale, Kilifi, and Lamu counties in Kenya. The admixture plot, a cross-validation of the cross-entropy value K = 2, indicates a pattern of similarity among cashew landraces from Kwale and Kilifi hence they share a genetic structure with an even distribution of two ancestral populations (red and blue clusters) for both markers. However, besides Lamu's landraces having a different genetic structure for SNP markers, it conforms to K = 2. The groupings of these individual cashew landraces within the admixture plot reflect the historical exchange of cashew seeds through informal systems or germplasm exchange in propagative breeding programs. Both the silicoDArT and SNP markers indicated two ancestral populations. The selection of K = 2 effectively captures these regional differences in genetic structure. Therefore, it can be hypothesized that human activities, such as using seeds from the same tree to grow new seedlings, could be a source of gene flow [37]. Our findings for genetic diversity and population structure in cashew employing ultrahigh-throughput DArTseq-based silicoDArT markers is fundamental towards understanding the progressive knowledge in cashew breeding and the conservation of germplasm similar to [26] work on cashew in Tanzania. The DArT platform allows simultaneous characterization of a substantial number of markers across the cashew genome, facilitating the efficient evaluation of its genetic diversity in breeding programs. Our AMOVA supports the moderate genetic diversity observed (Table 5), as indicated by allelic richness and the influence of cross-pollination, which is linked to cashew landraces outcrossing in this study.

Individual cashew landraces were clustered into three subpopulations, which correlated with their geographical locations. This observation supports the single domestication theory [38], highlighting a significant genetic bottleneck and limited genetic variation in cashew. This observation reiterates the single domestication theory which is a key genetic bottleneck and limited genetic variation in cashew. Understanding cashew genomics through the application of SilicoDArT and SNP markers is crucial for advancing breeding programs aimed at improving yield and nut quality, and enhancing resistance or tolerance to biotic and abiotic stresses, thereby optimizing selection outcomes.

## Conclusion

Our study evaluated silicoDArT and SNP markers and determined their usefulness in assessing the extent of genetic diversity and population structure among individual cashew landraces, as revealed in the cross-entropy and admixture plots. The genetic diversity among the cashew landraces for both silico DArT and SNP markers reflects a steady gene flow across the study sites which shows its adaptability to disease and climate change. However, the outcrossing effect presents an important research gap towards understanding cashew genomics in breeding programs. This information will guide genotype selection and subsequent breeding programs to develop new cashew germplasms to increase genetic gains for economic traits. Our study provides a snapshot of genetic diversity at a specific point in time. A longitudinal study tracking changes in genetic diversity over time could provide insights into the temporal dynamics of cashew populations.

## Supporting information

**S1 Table. Data dartseq.**
(CSV)

**S2 Table. Silico-DArT markers identified through the DArTseq process show the presence/absence of variation and SNP markers generated from the DArTseq process across the cashew genome.**
(CSV)

**S3 Table. Information on the different files generated and their raw data is provided in an Excel file containing the metadata.**
(XLSX)

## Acknowledgments

The authors are grateful to Pwani University for providing the laboratory space at the Pwani University Biosciences Research Centre (PUBReC) to perform this work.

## Author contributions

**Conceptualization:** Wilton Mwema Mbinda.

**Formal analysis:** Dennis Wamalabe Mukhebi, Diana Jepkoech Karan, Pauline Wambui Gachanja, Brenda Muthoni Kamau, Pauline Wangeci King'ori, Bicko Steve Juma.

**Funding acquisition:** Wilton Mwema Mbinda.

**Investigation:** Dennis Wamalabe Mukhebi, Diana Jepkoech Karan, Pauline Wambui Gachanja, Brenda Muthoni Kamau, Pauline Wangeci King'ori, Bicko Steve Juma.

**Methodology:** Dennis Wamalabe Mukhebi, Pauline Wambui Gachanja, Brenda Muthoni Kamau, Pauline Wangeci King'ori, Bicko Steve Juma.

**Project administration:** Wilton Mwema Mbinda.

**Resources:** Wilton Mwema Mbinda.

**Supervision:** Wilton Mwema Mbinda.

**Validation:** Wilton Mwema Mbinda.

**Writing – original draft:** Dennis Wamalabe Mukhebi, Diana Jepkoech Karan, Pauline Wambui Gachanja, Brenda Muthoni Kamau, Pauline Wangeci King'ori, Bicko Steve Juma.

**Writing – review & editing:** Wilton Mwema Mbinda.

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
