## [Decision Letter · Decision Letter 0]

13 Aug 2024

PONE-D-24-27021Genetic diversity and population structure of Kenyan cashew (Anacardium occidentale L) landraces using DArTseq-based silicoDArT and SNP markersPLOS ONE

Dear Dr. Mbinda,

Thank you for submitting your manuscript to PLOS ONE. After careful consideration, we feel that it has merit but does not fully meet PLOS ONE’s publication criteria as it currently stands. Therefore, we invite you to submit a revised version of the manuscript that addresses the points raised during the review process.

We look forward to receiving your revised manuscript.

Kind regards,

Karthik Kannan, Ph. D.,

Academic Editor

PLOS ONE

Journal Requirements:

3. Thank you for stating the following financial disclosure: This research was supported by the National Research Fund, Kenya (NRF/2/MMC/158). 

Additional Editor Comments:

Dear authors, kindly revise the manuscript thoroughly in accordance with the reviewer reports. 

<pre class="c-comments__text u-pre-wrap u-font-merriweather" style="font-family: inherit; font-size: 14px; margin-top: 0px; margin-bottom: 0px; padding: 0px; box-sizing: border-box; color: rgb(34, 34, 34); line-height: 26px; letter-spacing: 0.14px; background-color: rgb(227, 240, 244);">Reviewers' comments:Reviewer's Responses to Questions</pre>

**Comments to the Author**

1. Is the manuscript technically sound, and do the data support the conclusions?

Reviewer #1: Yes

Reviewer #2: Partly

Reviewer #3: Yes

2. Has the statistical analysis been performed appropriately and rigorously?

Reviewer #1: Yes

Reviewer #2: No

Reviewer #3: Yes

3. Have the authors made all data underlying the findings in their manuscript fully available?

Reviewer #1: No

Reviewer #2: No

Reviewer #3: Yes

4. Is the manuscript presented in an intelligible fashion and written in standard English?

Reviewer #1: No

Reviewer #2: No

Reviewer #3: Yes

5. Review Comments to the Author

**Reviewer #1:**  The manuscript on the ‘Genetic diversity and population structure of Kenyan cashew (Anacardium occidentale L) landraces using DArTseq-based silicoDArT and SNP markers’ has a strong scientific merit. The genetic resources from the study are important for breeding and conservation.  However, some of the results and discussion are still not very clear and need revision. The Authors should also consider a language revision.

The point-to-point comments are as follows;

P3, L2-3; please clarify the statement. We know that most trees provide the services mentioned in L1-2. Is there any specific feature exhibited by cashew to suggest any adaptive advantage of this species in providing the services mentioned?

P3, L3-5; this looks like a general statement when it comes to tree technologies so, please explain further how cashew specifically can support climate-smart technologies or climate change adaptation

P3, L9; add a coma after East Africa

P3, L15-16; the Authors mention that the flowers are self-pollinated and at the same time allogamous. This is very contradictory, please check

P4, L2; delete hew

P4, L12; add a comma after DaRT

P4, L13-17; revise statement for clarity

P5, L1-2; Split into two sentences for clarity, replace ‘over’ with ‘in’ not to be repetitive. Therefore, I suggest;  DArT is a popular platform for genomic research  because it produces many molecular markers over the entire genome in a short period. It is also straightforward to use and reasonably cost-effective (Deres and & Feyissa, 2022).

P5, L7 Please refer to this publication for your information; Discovery Of Novel Single Nucleotide Polymorphic (Snp) Markers For Genetic Mapping Of Cashew (*Anacardium Occidentale* . L)

P5, L16 replace ‘leaves’ with ‘leaf’

P5, L17 delete extra bracket from citation

P5, L18 Add a comma after strata

P5, L17-20 clarify on the proportional bit of the trees and the farmers. What were the sizes per population? This is important since it has a bearing on the allelic richness estimates.

P5, L16-23 needs to be streamlined to avoid repetitions

P7, L4, indexes or indices?

P7, L14 replace ‘plotted’ with ’generated’

P7, L22 add a comma after filtered

P8, L1 delete the p and unify the decimal places

P8, L4 the SNP-based expected heterozygosity range is 0.33?

P8, L7-9 I don’t see how the cross-entropy or scree plots identify the two clusters. These need further explanation.

P8, L14 I suppose the values in brackets are FST.

P8, L14 an FST of 0.005 or even 0.01 are not high to say there is divergence. It is even clear that clusters are not delineated based on these predefined sub-counties. What about differentiation between Kwale and Kalifu?

P8, L16-18 This sentence is not clear, please rewrite. Also, I don’t see any differentiation between Kilifi and Lamu given the very low FST. You need to test the significance of the clusters shown on Figure 4 especially given the very low FST values among these populations

P8, L16 PCoA not PCA

P8, L20-22 The roots of the dendrogram of for the genetic dissimilarity are not clear to decipher any relationships. I see four colour clusters on the dendrogram. Please clarify. Also clarify the 2 subpopulations based on K- analysis and 3 clusters based on PCoA and dendrogram.

P9, L10, change ‘differentiation’ to ‘delineation’ to reduce confusion

P9, L11 What are these ‘conventional molecular markers’?

P9, L15 Provide the average PIC. This, not the range would give a better picture of the level of heterozygosity

P9, L19, replace ‘between’ with ‘among’

P10, L1 please explain how domestication and distribution affect genetic diversity

P10, L6-13, the text is unclear and needs revision

P10, L14, the allelic richness should be discussed in view of population sizes.

P11, L3, in this case what is artificial hybridization? Are there occurrences of controlled crossing?

P11, L4-5 contradicts the results on P8, L13-14. Please reconcile

P11, L7 there is no genetic differentiation across clusters, so please clarify how selecting individuals in these clusters can increase genetic gain.

P11, L11 reconcile with the 2 subpopulations, on P8 L7-8

P12, L11 2, 3 or 4 subpopulations?

Citations need to be consistent, with DOIs

**Reviewer #2:**  Manuscript number: PONE-D-24-27021

Manuscript title: Genetic diversity and population structure of Kenyan cashew (Anacardium occidentale L) landraces using DArTseq-based silicoDArT and SNP markers

The study was conducted to analyse the genetic diversity and population structure of cashew plants from three counties namely Kwale, Kilifi and Lamu in Kenya using two types of makers: SilicoDArT and SNP markers. The study reported new findings in the cashew plants in Kenya using Diversity Array Technology. Referring to results, cashew landraces in Kenya are moderately divergent demonstrating that they are closely related thought they were separated into 2-3 sub-populations based on the population structure. Findings on genetic diversity and population structure should potentially be required for setting up breeding and conservation programs for cashew plant in Kenya. However, while, the study is very interesting and contribute to the body of science, a further thorough improvement is needed. Comments and suggestions were provided below:

Major issues:

1. Although the abstract was well written it needs to be improved since, no summary of how the study was conducted, and some quality control criteria presented do not match those presented in the body of text (call rate and reproducibility). In addition, there is no clear overall aim of the study was stated in both abstract and introduction. Results presented were not enough for readers to ensure that readers understand exactly what the investigators studied (only PIC ranges presented). The authors should also revise the conclusion and relate it to the topic of the manuscript.

2. The section of materials and methods was developed but it needs the improvement because it misses some important information: the sampling method was not clearly described, it lacks the names of sub-counties, and the number of cashew landraces (samples) corrected from each subcounty and each county. The distance used between samples (which can have a big impact on results), and the motivation behind the choice of limiting the study to three counties only should be provided. The table showing the cashew genotypes, sub-counties and counties of origins should be included to provide some information on samples and sampling procedure. The authors should clarify the use of PIC as genetic diversity index in the section of materials and methods and presented in results as the quality control parameter in order to avoid confusion. Authors also used Landscape and Ecological Associations (LEA) R package to analyse the genetic population structure of cashew landraces and sub-populations were validated using Principal Component Analysis (PCA). They said that subpopulations of cashew landraces for three counties were identified using a Q matrix in R statistical software and represented graphically by the Principal Coordinate Analysis (PCoA). This is another technique for analysis of population structure. Subsequently, they should clearly explain why they wanted to perform another population structure analysis. If it was a cross-validation what happen when the results are not validated as expected since the groups created by these techniques were different? Since only local cashew landraces were used, the admixture should be strong enough if some other cashew varieties (external for example as reference) were used to analyse whether the Kenyan landraces are really purely indigenous for the purpose of conservation or breeding.

3. The interpretation of results was not enough to ensure readers understand the main key components in the topic such as genetic diversity and population structure. They were not presented in logical order which can be difficult particularly who are not experts in discipline to understand how findings relate to these genetic components. For a good order, subheadings in this section should sound good. Also, some results presented in this section do not match those presented in Table 2. I advise authors to check and make corrections. The authors have missed to present in this section crucial results demonstrating the level of genetic diversity within populations. These findings (He and Ho) were also presented in no right subsection “markers’ characterisation” and are not the same as those presented in Table 2. The authors should also explain why no genetic indices of cashew landraces from Lamu were calculated. Findings revealed that the techniques used in the analysis of population structure of cashew landraces created different groups (clusters), however, the authors should clarify why the difference and which ones were validated and why? Or they should explain their complementarity. To ensure readers understand the relationship between cashew landraces from the three populations (Kwale, Kilifi and Lamu), the membership of cashew landraces should be indicated in two subpopulations formed based on cross-entropy and three subpopulations formed based on Q matrix and PCoA. E.g. the number of landraces from Kwale, Kilifi and Lamu assigned to cluster 1, etc. In addition, figures (Fig. 3 and Fig. 5) were blindly presented, only codes of cashew landraces were presented without corresponding populations, difficult to identify clusters 1, 2, 3 and their composition for both types of markers.

4. Although the section of discussion was good requires the improvement. The authors stated that the expected heterozygosity was greater than the observed heterozygosity in contrast of what presented in Table 2 (which misses the indices of Lamu population). This should reflect the implication of these genetic indices on the background and breeding history of cashew landraces already highlighted. I agree with authors that the allele richness among individuals may play a role in their ability to adapt to some stressors when is high, authors should justify thoroughly their confirmation without ambiguity that the moderate allele richness observed in cashew landraces explains their ability to adapt to both diseases and climate changes. Furthermore, the number of clusters formed by both markers is the same (3) but the composition of those clusters is different between two types of markers used. Authors should discuss and explain the reason behind the difference and the implication of these results on the applicability of either type of markers. Since the study looked at the usefulness of DArT markers in genomic studies, related findings and applicability were not discussed with other types of markers used in previous studies on genetic diversity in cashew plant and/or in other plants in the country or region to scrutinize their suitability and success. In addition, the discussion section needs to be improved and revised avoiding some confusion (e.g. the sentence on page 11 L 19 – 20 itself), contradictions (e.g. Page 11 L 13 – 16 and page 11 L 19 – 20), and explaining deeply findings with or against those that can be found in the literature. The division of this section into at least two subsections should sound better. Normally using seeds from the same tree can limit the genetic variation and favour the increase in inbreeding rather than the gene flow.

5. The conclusion section was good but it always needs the improvement. Conclusions should be drawn by also taking into account the key findings such as relationship between landraces (ancestral subpopulations and three subpopulations), admixture and the potential fixation of alleles revealed by the heterozygosity. In discussions, it was stated that “the use of a dual-marker approach, incorporating both silicoDArT and SNP markers elevates the precision and reliability of genetic diversity assessments capturing various types of variations within individual cashew populations”. Nevertheless, these markers showed the high divergence in population structure (difference in cluster composition). Authors should clarify the reason behind and the implication on the decision making. Authors should also highlight some limitations and suggest or propose the potential future research.

6. The manuscript didn’t follow the reference style and citation format. Tables and their captions were placed at the end of the manuscript.

7. The study has used two types of data: data for SNP markers and data for SilicoDArT markers. Referring to the metadata file, SNPs should have two formats:

- SNP 2 Rows Format: Each allele scored in a binary fashion ("1"=Presence and "0"=Absence). Heterozygotes are therefore scored as 1/1 (presence for both alleles/both rows) and

- SNP 1 Row Mapping Format: "0" = Reference allele homozygote, "1" = SNP allele homozygote, "2"= heterozygote and "-" = double null/null allele homozygote (absence of fragment with SNP in genomic representation)

while

For SilicoDArT Format: SilicoDArTs are scored in a binary fashion, representing genetically "dominant" markers, with "1"=Presence and "0"=Absence of a restriction fragment with the marker sequence in genomic representation of the sample. "-" represents calls with non-zero counts, but too low counts to score confidently as "1" (often representing heterozygotes)

The data for SNP markers was presented neither in 2 rows format nor in 1 row format. For that reason, both types of markers are presented in the same format. I addition, the data uploaded do not contain all necessary information characterizing the loci of Cashew landraces in Kenya such as chromosomes, positions on chromosomes, allele sequence, call rates, One ration, PIC, reproducibility, …. I suggest the authors should upload the correct and complete data that really support the findings and conclusions.

In summary, while methods used and findings can provide insightful information, the manuscript does't meet all requirements for publication. The main reasons behind are as follow: 1) it failed to be presented in the recommended structure and written in a comprehensive way to ensure readers understand the research. 2) The sampling method and analysis were not robust and clear. 3) Results presented were incomplete and mismatched. 4) The Tables were separated from the text, reference style and citation format were not respected. I suggest the manuscript be returned to authors with the option to resubmit it after substantial revisions. The authors should thoroughly address all the feedback provided, ensuring clarity, consistency, and adherence to journal guidelines. If the authors can make the necessary improvements, the study has the potential to make a significant contribution to the understanding of genetic diversity in Kenyan cashew landraces.

Minor issues:

Abstract

Page 1 L 14 – 21: this background is too long, it should be summarized to one sentence and focus on the aim and short description of the methods.

Page 1 L15: tree: replace “t” with “T”

Page 1 L 18: the authors indicated that several techniques have been employed to know the genetic diversity of different plants. It would be better they indicate whether there are some techniques used specifically for Cashew plant.

Page 2 L 2: PIC, ≥ 0.25 remove “, “

Page 2 L 4: remove “mean”

Page 2 L 4: remove “p” before 0.02?

Page 2 L 4: Replace “mean” with “average”, this should sound good.

Introduction

Page 3 L 3 – 5: the reference is missing

Page 4 L 5 – 6: it would sound good if this sentence (idea) is related to the previous idea L 1 – 3. Here the logic link between two ideas was lost (cashew & plants).

Materials and Methods

Page 5 L 17: reference format

Page 5 L 18 – 20: the sentence should be revised to make it clear.

Page 6 L 21: One ration, threshold???

Page 7 L 3 – 5: for clarity (since the aim was to determine the genetic diversity), the sentence should change because the genetic parameters presented were determined to indicate the level of genetic diversity.

Page 7: it would be worth if versions of tools used in analysis were provided

Results

Page 8 L 1: remove “p” before 0.02

Page 8 L 1: normally only ranges of PIC were presented rather than mean ranges of PIC. Referring to Table 1, the results are clear. Ranges should not be confused with means. Ranges would be presented along with means (but better to use “averages”) or medians.

P 8 L 2 – 3: this sentence should be revised

P 8 L 12: is there the difference between the subpopulations said on the same page L 8? If not, should the “subpopulation” in L 12 be changed into populations (original populations: Kilifi, Kwale and Lamu)?

P 8 L 16: principal coordinate analysis (replacing “PCA” with “PCoA”)

P 8 L 16: “showed” instead of show and put “that” after showed

P 8 L 16 – 18: the sentence should be revised. Low variance between Lamu and other counties and high variance within cashew landraces from Lamu do not explain the exception of Lamu county, since if you check the variance between cashew landraces from Kilifi and Kwale is also low and the variance within populations is high as shown by AMOVA.

P 8 L 20: colon (“:”) before “cluster 1”

P 8 L 21: “SilicoDArT” instead of “Silicon DArT”. And write “SilicoDArT markers” or “SilicoDArTs” and correct it throughout the document where it is needed

Discussions:

Page 11 L 6: add “s” at the end of population

Page 11 L 21 – 23 and Page 12 L 1 – 5: It is a part of conclusion rather than discussions

Figures

Fig. 1: Replacing “SilicoDArT reproducibility” by “Reproducibility of SilicoDArT markers”

Replacing “SNP reproducibility” by “Reproducibility of SNP markers”

Replacing “SilicoDArT data” by “Call rates of SilicoDArT markers”

Replacing “SNP call rate” by “Call rates of SNP markers”

Fig. 3: Legend

Figure 5: Legend

Format

Style and format: the line numbering was not continuous, it restarted for each page

Reference style: The manuscript didn’t follow the reference style for almost of references

**Reviewer #3:**  Here I have mentioned Detailed review comments for each section of the manuscript titled "Genetic diversity and population structure of Kenyan cashew (Anacardium occidentale L) using DArTseq-based silicoDArT and SNP markers":

Abstract

1. Key Findings: Mention the three distinct population groups identified in the SNP data analysis in more detail. It’s a significant result that should be highlighted.

Introduction

3. Background Information: The introduction provides a comprehensive background on cashew cultivation and its challenges. However, more context on the importance of genetic diversity studies in cashew could strengthen the rationale.

4. Current Knowledge Gaps: Clearly state the gaps in current research, specifically what this study addresses, to provide a better foundation for the objectives.

5. Objectives: The objectives are implicit. It would be helpful to explicitly state the primary goals of the study.

Materials and Methods

6. Sampling Strategy: The sampling strategy is described, but more detail on the stratified random system and why it was chosen would enhance clarity.

7. DNA Extraction: The DNA extraction method is standard, but mentioning any modifications or challenges encountered during the process could be useful for reproducibility.

8. DArTseq Genotyping: This section is detailed but could benefit from a flowchart or diagram to visually represent the workflow from DNA extraction to genotyping.

9. Statistical Analysis: The methods for statistical analysis are appropriate. However, providing more detail on the software versions and specific packages used in R would improve transparency.

Results

10. Data Presentation: The presentation of results is clear, but the inclusion of more figures, such as heatmaps or additional PCA plots, could help visualize the data better.

11. Marker Quality: The discussion on marker quality is adequate. Adding comparative analysis with other studies using similar markers would contextualize the findings.

12. Genetic Relationships: The results on genetic relationships and population structure are significant. However, discussing potential reasons for the observed admixture and gene flow in more detail would add depth.

Discussion

13. Interpretation of Findings: The discussion interprets the results well, but linking the findings to practical implications for breeding programs could be elaborated.

14. Comparative Analysis: Compare the results more extensively with similar studies in cashew or other crops to highlight the significance and novelty of the findings.

15. Limitations: Address the limitations of the study, such as the sample size or geographical constraints, and how they might affect the results.

Conclusion

16. Summary of Findings: The conclusion summarizes the findings effectively. However, suggesting specific future research directions based on the results would enhance this section.

17. Practical Implications: Highlight the practical implications for cashew breeding programs and conservation efforts more explicitly.

Figures and Tables

18. Figure Clarity: Ensure all figures are high resolution and clearly labeled. Figures 2 and 3, depicting population structure and admixture, should have clear legends and be easy to interpret.

19. Table Completeness: Tables should be self-explanatory with detailed captions. Ensure all abbreviations are defined in the table legends.

References

20. Citation: Add most recent references. Some references are more than a decade (2009, 2006, and 2001).

By addressing these comments, the manuscript can be improved for clarity, depth, and overall quality.

6. PLOS authors have the option to publish the peer review history of their article (what does this mean? ). If published, this will include your full peer review and any attached files.

**Do you want your identity to be public for this peer review?** For information about this choice, including consent withdrawal, please see our Privacy Policy .

Reviewer #1: No

Reviewer #2: No

Reviewer #3: No

---

## [Author Response · Author response to Decision Letter 1]

30 Aug 2024

We have responded to reviewer comments in the submitted responses to reviewers

---

## [Decision Letter · Decision Letter 1]

17 Sep 2024

PONE-D-24-27021R1DArTseq-based silicoDArT and SNP markers reveal the genetic diversity and population structure of Kenyan cashew (Anacardium occidentale L.) landracesPLOS ONE

Dear Dr. Mbinda,

Thank you for submitting your manuscript to PLOS ONE. After careful consideration, we feel that it has merit but does not fully meet PLOS ONE’s publication criteria as it currently stands. Therefore, we invite you to submit a revised version of the manuscript that addresses the points raised during the review process.

We look forward to receiving your revised manuscript.

Kind regards,

Karthik Kannan, Ph. D.,

Academic Editor

PLOS ONE

Journal Requirements:

**Additional Editor Comments:**

Dear Authors

The authors should revise the manuscript based on the reviewer 1 comments.

#Reviewer 1

DArTseq-based silicoDArT and SNP markers reveal the genetic diversity and population structure of Kenyan cashew (Anacardium occidentale L.) landraces

Dear Authors,

The manuscript has greatly improved. However, (i) the abstract needs to be more representative of the entire manuscript by adding more results such as the fixation indices, clustering, etc.

(ii) there the authors need to work on the flow, as some statements look very misplaced. (iii) There are also a number of scientific facts in the discussion that need to be checked.

The specific comments are presented below.

P2, L7-9, ‘Employing a dual-marker approach that combines SilicoDArT and SNP markers improves the precision and reliability of genetic diversity assessments by capturing different types of variation within individual cashew populations.’ I wonder how true this statement is especially that you are not comparing the performance of SilicoDArT and SNP markers with other markers, not even in the discussion. You may need to revise the statement.

L6 delete extra hyphens

P2, L23 may be ‘regional’ instead of ‘national’ economy since you are talking of a continent

P4, L8 delete comma

P4, L17 should it be ‘utilizes’ instead of utilize?

P5, L6 there is a publication of DarTseq in Cashew undertaken in Tanzania..” DISCOVERY OF NOVEL SINGLE NUCLEOTIDE POLYMORPHIC (SNP) MARKERS FOR GENETIC MAPPING OF CASHEW(Anacardium occidentale. L)”

P5, L17 ‘samples collected was proportional to the number of farmers and the density of cashew trees’. Please clarify for example, did you collect samples from each farmer that had cashew trees? Did you sample all the trees that individual farmers had or what does proportionality mean here?

P7, L9 delete extra hyphen

P7, L20 already presented in the methods, may delete

P8 L2-8. Results should be based on the filtered data only. I see an inconsistence for example in the PIC values reported at L3 and those at L4. Those at L4 seem to be summarised from the unfiltered data or if not, please clarify. Delete extra hyphens and throughout the manuscript (0.56--0.57)

P9, L21, please report the averages for both silicodarts and snps

P10, L1-2, please further explain how domestication relates to the genetic diversity. Ideally most domestication events are supposed to reduce diversity due to founder events and genetic bottlenecks associated with the sampling process.

P10, L7-10, heterozygote advantage may not be quite right here. Outcrossing is an OK explanation, and possibly there could have been more than one domestication events that caused mixing of two previously isolated populations as suggested on P11, L18-24.

P11, L7-8 The fixation indices are too low, and this should suggest high gene flow. Please revise the controversies here. The low fixation indices explain the lack of distinct population structure suggested in L9.

P11, L15-17, ‘This foundational work was crucial in advancing our understanding of genetic diversity and structure among 92 individual cashew landraces across Kwale, Kilifi, and Lamu Counties in Kenya’ doesn’t flow quite well here. I suggest you delete

P12, L3-8, don’t flow and need to be revised. You may move them to the concluding remarks.

P12,L12-14, could you please explain the 3 clusters vs K=2

Fig 4 and 5, Could you please color the different sub-populations differently for easy visualization of admixing.

# Reviewer 2

Accepted

Reviewers' comments:

Reviewer's Responses to Questions

**Comments to the Author**

1. If the authors have adequately addressed your comments raised in a previous round of review and you feel that this manuscript is now acceptable for publication, you may indicate that here to bypass the “Comments to the Author” section, enter your conflict of interest statement in the “Confidential to Editor” section, and submit your "Accept" recommendation.

Reviewer #1: (No Response)

Reviewer #2: All comments have been addressed

2. Is the manuscript technically sound, and do the data support the conclusions?

Reviewer #1: Yes

Reviewer #2: Yes

3. Has the statistical analysis been performed appropriately and rigorously?

Reviewer #1: Yes

Reviewer #2: Yes

4. Have the authors made all data underlying the findings in their manuscript fully available?

Reviewer #1: Yes

Reviewer #2: Yes

5. Is the manuscript presented in an intelligible fashion and written in standard English?

Reviewer #1: Yes

Reviewer #2: No

6. Review Comments to the Author

Reviewer #1: The presentation of the manuscript has greatly improved but still needs some work before publication. The topic and sound and standard data analyses have been applied.

Reviewer #2: Manuscript number: PONE-D-24-27021R1

Manuscript topic: DArTseq-based silicoDArT and SNP markers reveal the genetic diversity and population structure of Kenyan cashew (Anacardium occidentale L.) landraces

The manuscript is still interesting and may contribute to the body of science since findings on genetic diversity and population structure should potentially be required for setting up breeding and conservation programs for cashew plant in Kenya. However, I went through comments provided during the previous review and noticed that some were moderately addressed. Nevertheless, even though some were not addressed as expected, the manuscript is valid and can be published but after addressing the new comments for correction and improving its quality.

The additional comments are provided here below:

Abstract:

- The study aim is always missing or unclear. If authors look at it again would be better

- Results are still insufficient in the abstract. PIC ranges only cannot help to understand the conclusion drawn

- P 1 L 6: since they used ranges, the term “mean” should be removed

- P 1 L 6: replace “--" used in ranges by “to”

- The conclusion drawn in the abstract is not appropriate based on the results presented there.

In general, if the abstract is revised again it would be worth

Materials and Methods

- The sampling method was well described and the table inserted

- The motivation of choosing Kilifi, Kwale and Lamu counties was provided but not included inside the text of the manuscript. It would be worth if it is included.

- Different methods for determining the population structure were provided but the authors should explain the reason why each of them was used.

- P 7 L 2 – 5: this sentence is more confusing. Genetic indicators such PIC, allelic richness, He, Ho, … are determined in order to show the level of genetic diversity instead of conducting genetic analysis to determine the genetic indicators. I advise the authors to change this sentence. I thought they conducted the genetic analysis by determined these genetic indices.

Results

Comments were addressed. However, for improving the quality of the manuscript some corrections are still needed:

- P 7 L 20: the information here is not correct. Referring to Figure 1D, only 73.45% of SNPs presented a call rate ≥ 0.80, not all markers as it was said. The authors should interpret results by checking whether information in the text really matches what was presented in figures or tables.

- P 8 L 3: I was not able to catch well these ranges of PIC values when observing Figure 2.

- P 8 L2: replace “silicoDArt” by “SilicoDArTs”

- P 8 L4: harmonize along the text how silicoDarT is written. Normally is written as follows: “SilicoDArT markers” or “SilicoDArTs”

- P 8 L 4: remove the term “mean” since ranges were used. Similarly, replace “--" within ranges by “to”.

- P 8 L 7 - 8: replace “--" within ranges by “to”.

- P 8 L 8: replace “the” by “for”

- P 8 L 6 - 8: Since the heterozygosity in the key index of the genetic diversity, referring to He and/or Ho and Table 3, authors should describe which cashew population (s) revealed the high/low genetic diversity.

- P 9 L1: are LM/LW/106, LM/LW/88 and LM/LE/114 SilicoDArT markers or cashew landraces? I advise authors to check and correct.

- P 8 L 21-22: incorrect citation of figure (Fig. 7A). Authors should check if necessary whether this figure described the neighbour joining tree of SNPs.

- The title of table 2 should change, since the current title causes confusion. Maybe, it should change to “The distribution of PIC values for both SilicoDArT and SNP markers”. It would be better if authors check for all other titles and subtitles throughout the manuscript.

Discussions

All comments were addressed. However:

- P 9 L 10-12: this sentence should be revised and the reference (s) was (were) required.

- P 12 L 1-2: Authors should think on this statement or hypothesis. The seeds from the same tree or from related trees cannot favour the gene flow or genetic diversity. But the fixation of alleles (or homozygosity) may increase. This sentence is more confusing.

- P 11 L 17-24 and P 12 L 1: authors should check the font and size of the text and change it if necessary

Conclusions

Comments were moderately addressed. But so far, readers shall not be able to know whether these markers are equally suitable for genetic diversity and population structure in Kenya since even though they provided the same number of groups, the structure (clusters ‘composition) is widely different between both types of markers.

7. PLOS authors have the option to publish the peer review history of their article (what does this mean? ). If published, this will include your full peer review and any attached files.

**Do you want your identity to be public for this peer review?** For information about this choice, including consent withdrawal, please see our Privacy Policy .

Reviewer #1: No

Reviewer #2: No

---

## [Author Response · Author response to Decision Letter 2]

24 Sep 2024

Authors have responded to all Reviewer 1 commends and revised the manuscript accordingly. Where not possible, we have given a rebuttal.

---

## [Decision Letter · Decision Letter 2]

6 Oct 2024

PONE-D-24-27021R2DArTseq-based silicoDArT and SNP markers reveal the genetic diversity and population structure of Kenyan cashew (Anacardium occidentale L.) landracesPLOS ONE

Dear Dr. Mbinda,

Thank you for submitting your manuscript to PLOS ONE. After careful consideration, we feel that it has merit but does not fully meet PLOS ONE’s publication criteria as it currently stands. Therefore, we invite you to submit a revised version of the manuscript that addresses the points raised during the review process.

We look forward to receiving your revised manuscript.

Kind regards,

Karthik Kannan, Ph. D.,

Academic Editor

PLOS ONE

Journal Requirements:

Additional Editor Comments:

The authors need to revise the manuscript based on the reviewer 1 comments (see the attachment).

Reviewers' comments:

Reviewer's Responses to Questions

**Comments to the Author**

1. If the authors have adequately addressed your comments raised in a previous round of review and you feel that this manuscript is now acceptable for publication, you may indicate that here to bypass the “Comments to the Author” section, enter your conflict of interest statement in the “Confidential to Editor” section, and submit your "Accept" recommendation.

Reviewer #1: All comments have been addressed

2. Is the manuscript technically sound, and do the data support the conclusions?

Reviewer #1: Yes

3. Has the statistical analysis been performed appropriately and rigorously?

Reviewer #1: Yes

4. Have the authors made all data underlying the findings in their manuscript fully available?

Reviewer #1: Yes

5. Is the manuscript presented in an intelligible fashion and written in standard English?

Reviewer #1: Yes

6. Review Comments to the Author

Reviewer #1: The manuscript has greatly improved. I would recommend that it is accepted although some (mostly grammatical) issues still need to be addressed. The authors are advised to review the grammar. The citations also need to be consistent.

7. PLOS authors have the option to publish the peer review history of their article (what does this mean? ). If published, this will include your full peer review and any attached files.

**Do you want your identity to be public for this peer review?** For information about this choice, including consent withdrawal, please see our Privacy Policy .

Reviewer #1: No

---

## [Author Response · Author response to Decision Letter 3]

7 Oct 2024

Authors have revised the manuscript as per the commends made by the reviewer.

---

## [Decision Letter · Decision Letter 3]

1 Nov 2024

DArTseq-based silicoDArT and SNP markers reveal the genetic diversity and population structure of Kenyan cashew (Anacardium occidentale L.) landraces

PONE-D-24-27021R3

Dear Dr. Mbinda,

We’re pleased to inform you that your manuscript has been judged scientifically suitable for publication and will be formally accepted for publication once it meets all outstanding technical requirements.

Kind regards,

Karthik Kannan, Ph. D.,

Academic Editor

PLOS ONE

Reviewers' comments:

Reviewer's Responses to Questions

**Comments to the Author**

1. If the authors have adequately addressed your comments raised in a previous round of review and you feel that this manuscript is now acceptable for publication, you may indicate that here to bypass the “Comments to the Author” section, enter your conflict of interest statement in the “Confidential to Editor” section, and submit your "Accept" recommendation.

Reviewer #1: (No Response)

2. Is the manuscript technically sound, and do the data support the conclusions?

Reviewer #1: Yes

3. Has the statistical analysis been performed appropriately and rigorously?

Reviewer #1: Yes

4. Have the authors made all data underlying the findings in their manuscript fully available?

Reviewer #1: Yes

5. Is the manuscript presented in an intelligible fashion and written in standard English?

Reviewer #1: Yes

6. Review Comments to the Author

Reviewer #1: Dear Editor,

The manuscript has tremendously improved.

7. PLOS authors have the option to publish the peer review history of their article (what does this mean? ). If published, this will include your full peer review and any attached files.

**Do you want your identity to be public for this peer review?** For information about this choice, including consent withdrawal, please see our Privacy Policy .

Reviewer #1: No

---

## [Editor Report · Acceptance letter]

PONE-D-24-27021R3

PLOS ONE

Dear Dr. Mbinda,

I'm pleased to inform you that your manuscript has been deemed suitable for publication in PLOS ONE. Congratulations! Your manuscript is now being handed over to our production team.

Kind regards,

on behalf of

Prof. Karthik Kannan

Academic Editor

PLOS ONE